# Daptomycin forms a stable complex with phosphatidylglycerol for selective uptake to bacterial membrane

Pragyansree Machhua, Vignesh Gopalakrishnan Unnithan, Yu Liu, Yiping Jiang, Lingfeng Zhang, Zhihong Guo*

Shenzhen Research Institute and Department of Chemistry, The Hong Kong University of Science and Technology, Hong Kong, China

## eLife Assessment

This **valuable** study describes the molecular mechanism of daptomycin insertion into bacterial membranes. The authors provide **solid** in vitro evidence for the early events of daptomycin interaction with phospholipid headgroups and stronger, specific interaction with phosphatidylglycerol. This work will be of interest to bacterial membrane biologists and biochemists working in the antimicrobial resistance field.

*For correspondence:
chguo@ust.hk

Competing interest: The authors declare that no competing interests exist.

**Abstract** Daptomycin is a potent lipopeptide antibiotic used in the treatment of life-threatening Gram-positive infections, but the molecular mechanism of its interaction with bacterial membrane remains unclear. Here, we show that this interaction is divided into two stages, of which the first is a fast and reversible binding of the drug to phospholipid membrane in milliseconds, and the second is a slow and irreversible insertion into membrane in minutes, only in the presence of the bacteria-specific lipid phosphatidylglycerol, to a saturating point where the ratio of the drug to phosphatidylglycerol is 1:2. Fluorescence-based titration showed that the antibiotic simultaneously binds two molecules of phosphatidylglycerol with a nanomolar binding affinity in the presence of calcium ion. The resulting stable complex is easily formed in a test tube and readily isolated from the membrane of drug-treated bacterial cells, strongly supporting a unique drug uptake mechanism in which daptomycin forms a stable multicomponent complex with calcium and phosphatidylglycerol. Revelation of this novel uptake mechanism provides fresh insights into the mode of action of daptomycin and paves the way to new strategies to attenuate resistance to the drug.

## Introduction

Daptomycin (Dap) is a calcium-dependent lipopeptide antibiotic used to treat life-threatening Gram-positive infections (*Liu et al., 2011*; *Baddour et al., 2015*). It exhibits potent bactericidal activity (*Mascio et al., 2007*; *Cotroneo et al., 2008*) and was once used as a last-line-of-defense antibiotic against resistant pathogens such as methicillin-resistant *Staphylococcus aureus* and vancomycin-resistant enterococci (*WHO, 2019*). Resistance to Dap is mostly mild with moderate increases in the minimum inhibitory concentration (*Sauermann et al., 2008*). However, recent years have witnessed an increasing number of failed treatments due to high-level Dap resistance of clinical pathogens, including *streptococci* (*Akins et al., 2015*; *García-de-la-Mària et al., 2013*), *Enterococcus faecium* (*Humphries et al., 2012*), and *Corynebacterium striatum* (*McElvania TeKippe et al., 2014*). Currently, few Dap derivatives are available to attenuate the resistance (*Chow et al., 2020*).

The effort to counter the increasing resistance is hampered by our insufficient understanding of the action mechanism of Dap, which has been suggested to involve the inhibition of peptidoglycan biosynthesis (*Allen et al., 1987*; *Mengin-Lecreulx et al., 1990*) or membrane depolarization (*Silverman et al., 2003*; *Zhang et al., 2014*). Recent studies have provided supporting evidence for the inhibition of peptidoglycan biosynthesis either by disruption of membrane fluidity (*Müller et al., 2016*) or by the formation of a bactericidal complex between the drug and lipid II (*Grein et al., 2020*). Another unresolved problem is how Dap recognizes Gram-positive bacteria and selectively accumulates in their membrane. Cumulative evidence has shown that this cell specificity is linked to phosphatidylglycerol (PG), a major phospholipid in most bacteria but rare in mammalian cells (*Taylor and Palmer, 2016*). PG was first linked to susceptibility to Dap by the finding that the antibiotic accumulates in membrane microdomains rich in this lipid (*Hachmann et al., 2009*). In addition, PG in pulmonary surfactants is able to neutralize Dap, rendering the drug ineffective against community-acquired, bacteria-caused pneumonia (*Silverman et al., 2005*; *Pertel et al., 2008*). More importantly, decreased membrane content of PG caused by reduction-of-function mutations in the PG synthase PgsA has been unambiguously shown to be primarily responsible for resistance to Dap, first in the model bacterium *Bacillus subtilis* (*Hachmann et al., 2011*) and then in clinical pathogens (*Hines et al., 2017*). When PgsA is inactivated by a single loss-of-function mutation, Dap susceptibility is completely eliminated to result in a very high level of resistance (*Goldner et al., 2018*). Corroborated by in vitro liposome model studies (*Goldner et al., 2018*), these findings strongly support a critical role of PG in the recognition of bacterial cells by the antibiotic.

Previous studies have shown that Dap nonspecifically binds membrane regardless of the lipid composition (*Taylor and Palmer, 2016*) and also interacts specifically with PG in membrane with conformational change (*Jung et al., 2004*; *Zhang et al., 2013*; *Lee et al., 2017*). However, the strength of this specific interaction and how it contributes to bacteria-specific uptake of the drug remain unclear. In addition, these early investigations reached different conclusions on the stoichiometry of the Dap interaction with PG (*Zhang et al., 2013*; *Lee et al., 2017*). In this study, we investigated the calcium-dependent interaction of Dap with model membranes and found that the mode of interaction is dependent on PG. In the absence of PG, Dap reversibly binds to the membrane in a fast process, whereas the drug undergoes a slow, irreversible insertion into the membrane when PG is present. Further investigation showed that Dap forms a multicomponent complex with calcium and two molecules of PG both in vitro and in drug-treated bacterial cells, thus revealing a unique mechanism for the selective uptake of Dap into bacterial membrane.

## Results

### Dependence of Dap uptake on PG

Interaction of Dap with the membrane is known to increase the fluorescence of its kynurenine residue (Kyn-13, *Figure 1*) with a 15 nm blueshift (*Lakey and Ptak, 1988*). We used the kinetics of this fluorescent increase to study the interaction in micelles and found negligible fluorescence increase when the model membrane was comprised of 1,2-dimyristoyl-*sn*-glycero-3-phosphocholine (DMPC) only (*Figure 2A*). When 1,2-dimyristoyl-*sn*-glycero-3-phosphorylglycerol (DMPG) was added to the phospholipid micelles, the fluorescence increased and reached a plateau in about 10 min. Interestingly, the plateau level is limited by the DMPG content and in most cases is unable to reach a maximum level at which all Dap is absorbed into the model membrane. In addition, the initial rate of the fluorescence change sharply increases with the concentration of both DMPG and $Ca^{2+}$ (*Figure 2—figure supplement 1*). Similar kinetics of Kyn fluorescence was observed in the interaction of Dap with vesicles containing DMPG (*Figure 2—figure supplement 2*), indicating that Dap interacts with PG-containing membrane in the same mode regardless of the model system.

Using PG-free micelles, no increase in fluorescence intensity was detected when the model membrane contained the same molar equivalent of other major lipid components in bacteria, including 1-palmitoyl-2-oleoyl-*sn*-glycero-3-phosphoethanolamine (POPE), cardiolipin (CL), 1-palmitoyl-2-oleoyl-*sn*-glycero-3-phospho-L-serine (POPS), 1-palmitoyl-2-oleoyl-*sn*-glycero-3-phosphate (POPA), and farnesyl pyrophosphate ammonium salt (FPP) (a structural homolog of bactoprenol diphosphate) (*Figure 2B*). Nevertheless, the steady-state Dap fluorescence also increases with the content of the lipids, particularly for the negatively charged POPS and CL. Noticeably, this enhanced

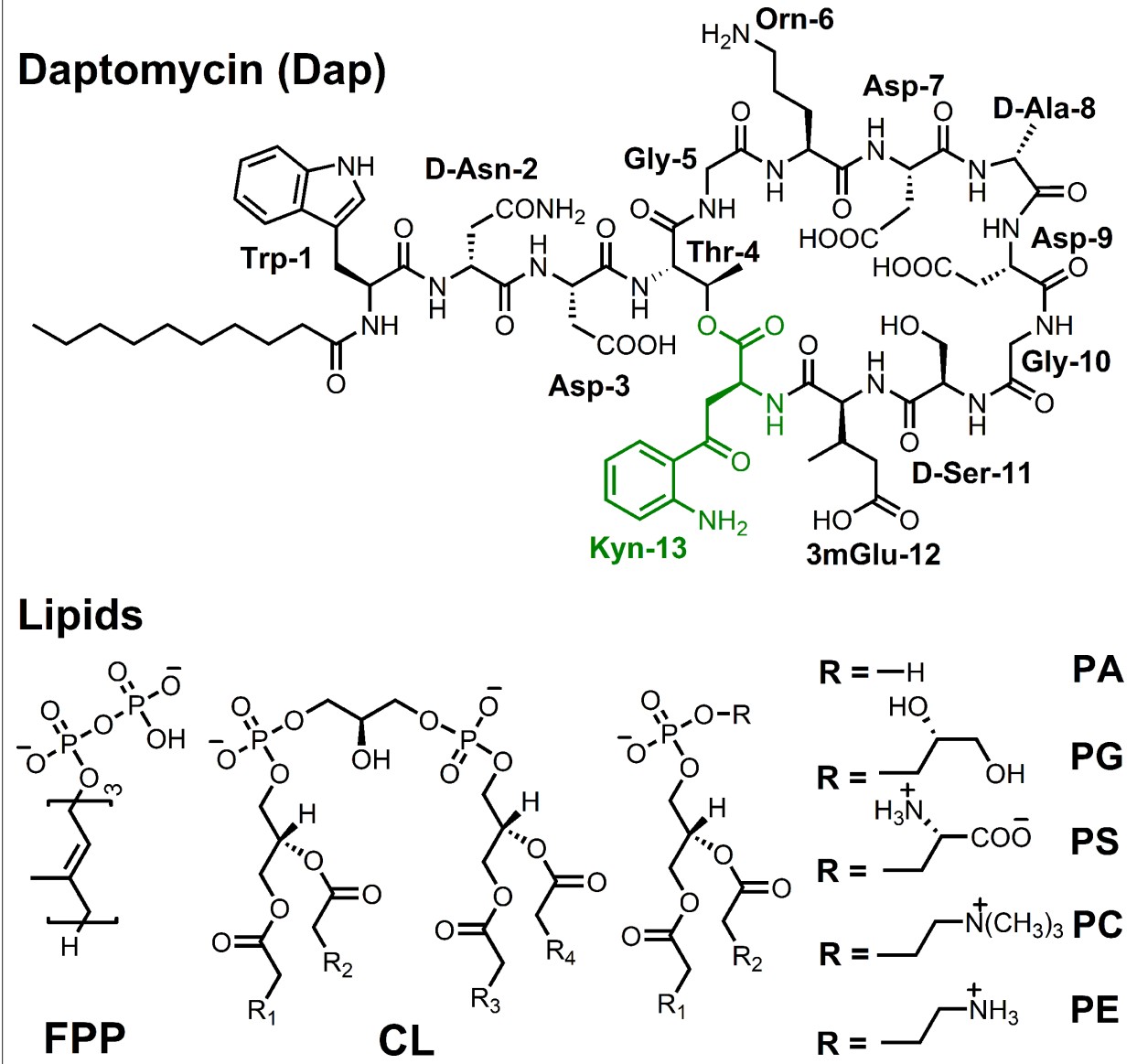

**Figure 1.** Structure of daptomycin and lipids.

background fluorescence is accompanied by the same blueshift observed in micelles containing PG (*Figure 2—figure supplement 3*), suggesting that Dap also interacts with the PG-free membrane, but in a mode different from that for the PG-containing membrane. Corroboratively, the steady-state Dap fluorescence is linearly proportional to the DMPG content and reaches a maximum at a concentration about twice that of Dap, but it has a significantly different relationship with the content of POPS, CL, or POPA (*Figure 2C*). PG-dependent increase of the steady-state fluorescence was also observed in giant unilamellar vesicles (*Krok et al., 2023*).

### Two distinct modes of Dap interaction with membrane

In the interaction of Dap with micelles containing CL, the increase of fluorescence was found to be complete within 100 ms by stopped-flow kinetics (*Figure 2D*), demonstrating that this mode of interaction occurs much faster than the slow accumulation of Dap in PG-containing membrane (over minutes, *Figure 2A*). This fast process is believed to correspond to the Ca²⁺-dependent recruitment of Dap, via diffusion, to the headgroup region of the phospholipid membrane, where the environment of the Kyn residue becomes more hydrophobic, likely due to structural change to cause the Kyn

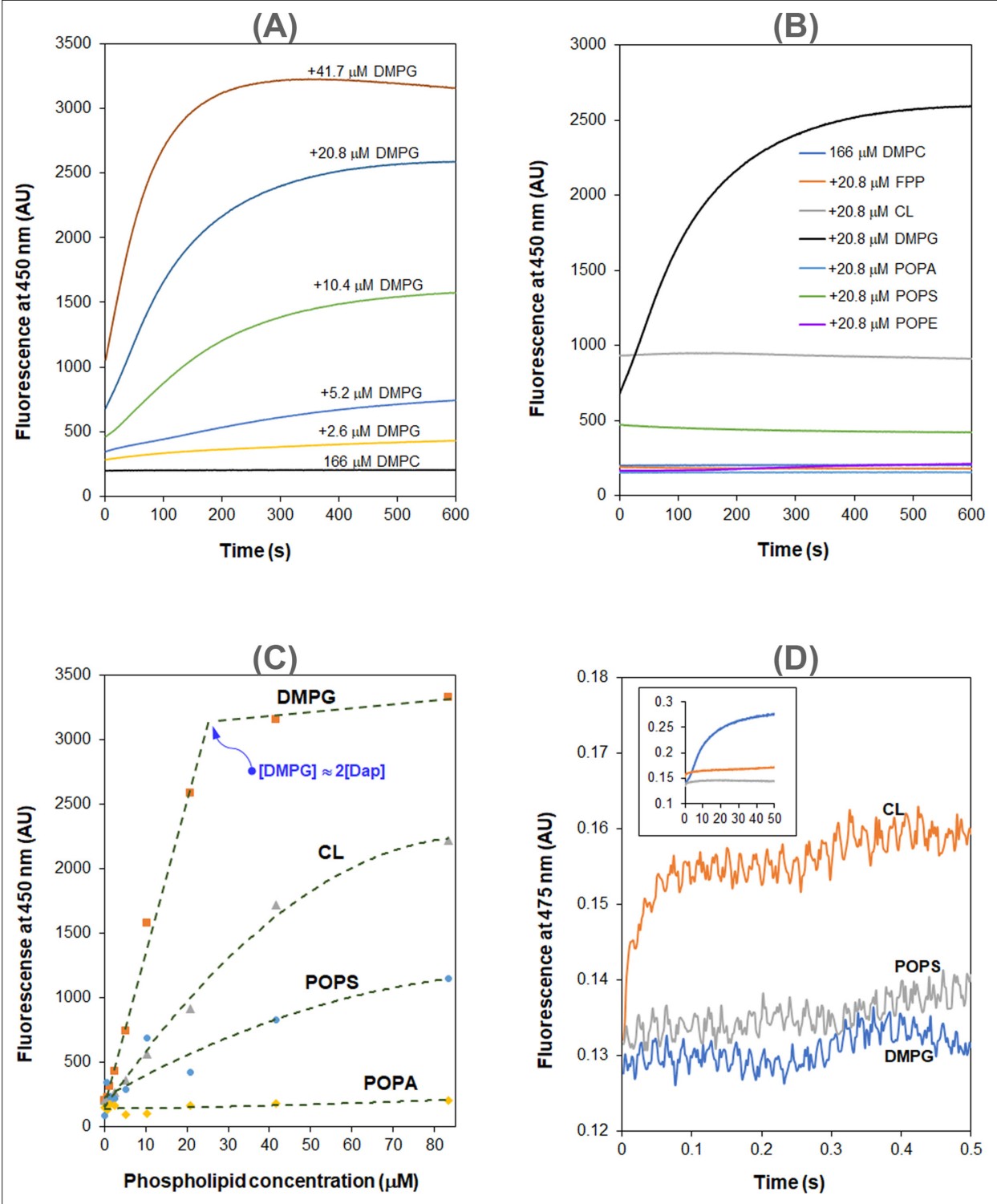

**Figure 2.** Daptomycin (Dap) depends on phosphatidylglycerol (PG) in the interaction with membrane. (**A**) 1,2-Dimyristoyl-*sn*-glycero-3-phosphorylglycerol (DMPG)-dependent accumulation of Dap in phospholipid micelles. (**B**) Kinetics of the fluorescence of Dap in interaction with micelles containing different phospholipids. Micelles contained 20.8 µM various phospholipids (11.5%) and 167 µM 1,2-dimyristoyl-*sn*-glycero-3-phosphocholine (DMPC). (**C**) Plot of the steady-state Kyn fluorescence vs. the content of different negative phospholipids in the micelles. (**D**) A fast Kyn fluorescence increase within 100 ms in the interaction of Dap with cardiolipin (CL)-containing micelles. The inset shows the Kyn fluorescence change over 50 s; micelles contained DMPG, 1-palmitoyl-2-oleoyl-*sn*-glycero-3-phospho-L-serine (POPS), or CL at 20.8 µM. All the experiments in (**A–D**) were

*Figure 2 continued on next page*

*Figure 2 continued*

carried out in 20 mM HEPES (pH 7.57) containing 1.67 mM CaCl₂; Dap was always added at last at 15 μM. The micelles were prepared from DMPC at 167 μM alone or together with another lipid at a concentration or content indicated in the plots.

The online version of this article includes the following source data and figure supplement(s) for figure 2:

**Source data 1.** Kyn fluorescence change with time in the interaction of Dap with DMPC micelles containing a varied amount of DMPG.

**Source data 2.** Kyn fluorescence change with time in the interaction of Dap with DMPC micelles containing a fixed amount of a phospholipid other than POPC or FPP.

**Source data 3.** Kyn fluorescence change with time in the interaction of Dap with DMPC micelles containing a varied amount of CL.

**Source data 4.** Kyn fluorescence change with time in the interaction of Dap with DMPC micelles containing a varied amount of POPS.

**Source data 5.** Stopped-flow kinetics of Kyn fluorescence change in the interaction of Dap with DMPC micelles containing the same amount of DMPG, CL, or POPS.

**Figure supplement 1.** Dependence of the daptomycin (Dap) uptake on both 1,2-dimyristoyl-*sn*-glycero-3-phosphorylglycerol (DMPG) and calcium.

**Figure supplement 1—source data 1.** The initial rate of Kyn fluorescence change increases with the DMPG content in DMPC micelles interacting with Dap.

**Figure supplement 1—source data 2.** The initial rate of Kyn fluorescence change increases with the calcium concentration in Dap interaction with DMPG-containing DMPC micelles.

**Figure supplement 2.** Enhancing effect of phosphatidyl glycerol (PG) on the daptomycin (Dap) uptake to vesicles.

**Figure supplement 2—source data 1.** Kyn fluorescence change with time in Dap interaction with DMPC vesicles containing a varied amount of DMPG.

**Figure supplement 2—source data 2.** The initial rate of Kyn fluorescence change increases with the DMPG content in Dap interaction with DMPC vesicles.

**Figure supplement 3.** Fluorescence enhancement and blueshift caused by negative cardiolipin (CL) and phosphatidyserine (PS).

**Figure supplement 3—source data 1.** Kyn fluorescence change with time in Dap interaction with DMPC micelles containg a varied amount of CL.

**Figure supplement 3—source data 2.** Kyn fluorescence change with time in Dap interaction with DMPC micelles containg a varied amount of POPS.

**Figure supplement 3—source data 3.** Emission spectra of Kyn in the interaction of Dap with DMPC micelles containg a varied amount of DMPG.

**Figure supplement 3—source data 4.** Emission spectra of Kyn in the interaction of Dap with DMPC micelles containg a varied amount of CL.

**Figure supplement 3—source data 5.** Emission spectra of Kyn in the interaction of Dap with DMPC micelles containg a varied amount of POPS.

fluorescence enhancement and blueshift. This binding to the membrane surface should be reversible, and no Dap is irreversibly inserted into the hydrophobic acyl layer of the membrane, since the Kyn fluorescence was not further increased afterward. Indeed, when $Ca^{2+}$ was sequestered by EGTA, the bound Dap was completely released back into the bulk solution, as indicated by the decrease of the Kyn fluorescence to the same level of free Dap control without the lipids (*Figure 3A*). In support of this reversibility, the dramatic conformational change identified for Dap bound to the CL micelles was completely reversed to be identical to free Dap after $Ca^{2+}$ sequestration (*Figure 3B*).

By contrast, Dap was inserted irreversibly into the DMPG-containing membrane, because the Kyn fluorescence continuously increased (*Figure 2A*) until all Dap was exhausted when the DMPG content was high enough (*Figure 2C*). This is supported by the substantial residual Dap fluorescence after EGTA sequestration of $Ca^{2+}$ from the accumulated drug in DMPG-containing micelles (*Figure 3A*). This residual fluorescence not only indicates that Dap remains bound to the membrane after removal of the metal ion but also demonstrates that Dap is still associated with $Ca^{2+}$ after insertion into the membrane. The fluorescence decrease induced by $Ca^{2+}$ sequestration suggests that Dap is buried deeper into the membrane when bound by the metal ion. The irreversible insertion of Dap into DMPG-containing micelles is further supported by the conformation of Dap after $Ca^{2+}$ sequestration, which remains significantly different from free Dap, albeit with significant change (*Figure 3B*). Interestingly, Dap shows a nonidentical, similarly shaped circular dichroism (CD) spectrum in its reversible and irreversible binding with the CL and DMPG micelles (*Figure 3B*), respectively, indicating a small conformational change in the insertion of the surface-bound Dap into the hydrophobic acyl layer of the membrane.

## $Ca^{2+}$-dependent interaction between Dap and DMPG

To assess whether the putative PG headgroup was responsible for the observed irreversible uptake of Dap, G3P was included in the kinetic measurement and found to slightly inhibit the drug uptake

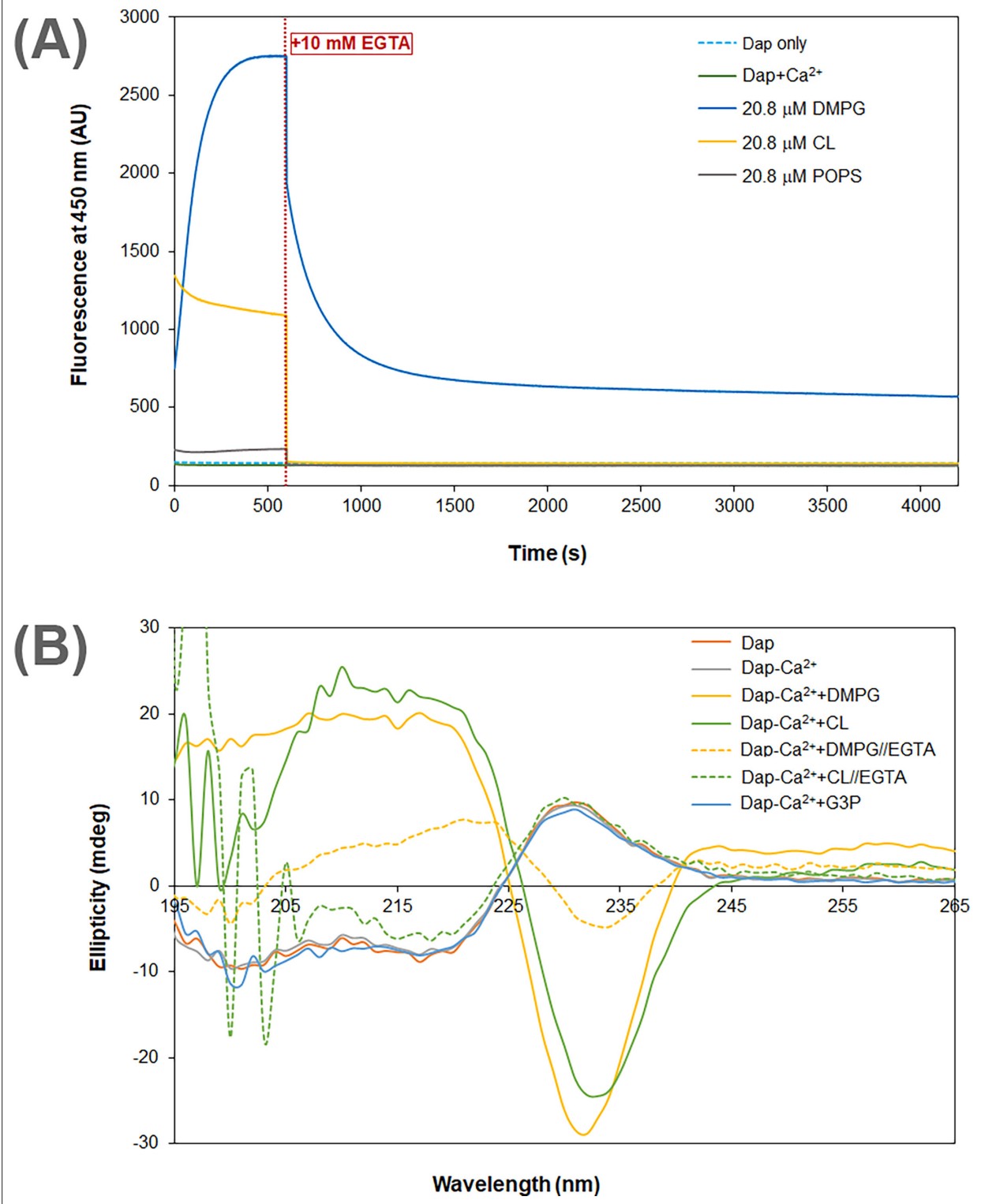

**Figure 3.** Calcium dependence of the daptomycin (Dap) structure and fluorescence in membrane. (**A**) Calcium sequestration decreases the Dap fluorescence in a way dependent on the phospholipid composition. The kinetic fluorescence measurement was performed under the same conditions as in *Figure 1B*. The red dotted line denoted the time point when 10 mM EGTA was added. The controls contained Dap at 15 µM Dap in the same buffer with or without calcium. (**B**) Circular dichroism spectra of Dap in different membrane environments. The buffer was 20 mM Tris·HCl, pH 7.57; Dap was 180 µM and Ca(CH$_3$COO)$_2$ was 1.0 mM; spectra were recorded 30 min after mixing with micelles contained pure 1,2-dimyristoyl-*sn*-glycero-3-phosphorylglycerol (DMPG) or cardiolipin (CL) at 720 µM to maximize the amount of bound Dap; in Ca$^{2+}$ sequestration experiments, spectra were

*Figure 3 continued on next page*

*Figure 3 continued*

recorded 30 min after adding 10.0 mM EGTA. The putative phosphatidylglycerol (PG) headgroup, *sn*-glycerol 3-phosphate (G3P), was 20 mM in an attempt to detect its potential interaction with Dap.

The online version of this article includes the following source data and figure supplement(s) for figure 3:

**Source data 1.** Kyn fluorescence change with time before and after adding 10 mM EGTA to the system for the interaction of Dap with DMPC micelles containing the same amount of DMPG, CL, or POPS.

**Source data 2.** Circular dichroism spectra of Dap in the interaction with DMPC micelles under various conditions.

**Figure supplement 1.** Interaction of daptomycin (Dap)-$Ca^{2+}$ with glycerol-3-phosphate (G3P).

**Figure supplement 1—source data 1.** Kyn fluorescence change with time in the interaction of Dap with DMPG-containing micelles in the presence of sn-glycerol-3-phosphate (G3P) at a varied concentration.

**Figure supplement 1—source data 2.** Kyn fluorescence change with temperature in the interaction of Dap with DMPG-containing micelles in the presence and absence of sn-glycerol-3-phosphate (G3P).

**Figure supplement 2.** Highly similar $N_\alpha H$-$H_\alpha$-$H_\beta$-$H_\gamma$ spin systems in daptomycin (Dap) at pH 5.0 and pH 5.40.

at millimolar levels (*Figure 3—figure supplement 1A*). This interaction was not detectable by CD (*Figure 3B*), isothermal titration calorimetry (ITC), or ligand-induced fluorescence change but was found to cause a small fluorescence-based thermal shift in the presence of 20 mM G3P (*Figure 3— figure supplement 1B*). In the NMR titration of G3P into a Dap-$Ca^{2+}$ solution at pH 5.40, the α-protons of Trp-1, D-Asn-2, Asp-9, and Kyn-13 were found to shift slightly in the fingerprint region (*Figure 3— figure supplement 1C*). The α-proton signals were assigned by comparing the unique $N_\alpha H$-$H_\alpha$-$H_\beta$-$H_\gamma$ cross-peak patterns of the amino acid residues from Dap in TOCSY with those obtained for the drug at pH 5.0 in a previous study (*Ball et al., 2004*) (see *Figure 3—figure supplement 2* for the comparison and *Supplementary file 1* for the assignments). These results show that Dap indeed binds G3P, but the affinity is too low (in high millimolar range) to account for the DMPG-enabled membrane insertion.

Noticeably, Dap was saturated by DMPG at a ratio of 1:2 in the steady-state fluorescence titration (*Figure 2C*), indicating a binding interaction between the two in 1:2 stoichiometry that is consistent with the previous ITC titration (*Zhang et al., 2013*). To determine the binding affinity, Dap at 15 nM was titrated with sub-micromolar DMPG in the presence of $Ca^{2+}$, and its Kyn fluorescence at 454 nm was found to increase sharply at low DMPG concentration and reach a saturating level at high concentration, whereas no fluorescence increase was observed for a control titration with POPS at an elevated Dap concentration of 1.5 µM (*Figure 4*). The titration curve was best fitted with a model in which Dap binds two molecules of DMPG with a dissociation constant of $K_D = 7.2 \times 10^{-15}$ $M^2$. DMPG has a critical micelle concentration (CMC) higher than 1.0 µM (*Figure 4—figure supplement 1*) and is homogeneously present in solution in the 0–1.0 µM concentration range under the given conditions. This titration result shows that Dap forms a high-affinity complex with two molecules of DMPG in the presence of calcium ion.

To obtain the complex, Dap (20 µM or higher) and DMPG were mixed at a 1:2 molar ratio in the presence of $CaCl_2$ (1 mM or higher), and white precipitate was immediately formed, while mixing DMPG with $CaCl_2$ resulted in a clear solution (*Figure 4—figure supplement 2*). By comparison, precipitate was formed when POPS or POPA was mixed with $CaCl_2$ only, whereas no precipitate was observed when Dap was mixed with other lipids under the same conditions. This comparison suggested that the precipitate was formed from the specific calcium-dependent binding interaction between Dap and DMPG. The same precipitate was also formed by extracting the DMPG-containing micelles after the Dap uptake experiments (*Figure 2A*) with chloroform, which was not observed in similar extraction of Dap-bound micelles without DMPG. This precipitate dissolved readily in aqueous solution containing 10 mM EGTA and was moderately soluble in a few organic solvents such as dimethyl sulfoxide (DMSO) and dichloromethane. In high-performance liquid chromatography (HPLC), the dissolved complex was a pure single species in the elution profile (*Figure 5A*).

## Dap-PG complex isolated from drug-treated bacterial cells

To test whether Dap is complexed with PG in cells, *B. subtilis* subsp. *subtilis* 168 was grown in Luria broth, treated with Dap at 0.375 mg/ml and $CaCl_2$ at 6.8 mM, and lysed after washing. The cell membrane was isolated from the crude extract by ultracentrifugation and extracted with chloroform. After adding sodium dodecyl sulfate (SDS) to solubilize all precipitated proteins, white precipitate

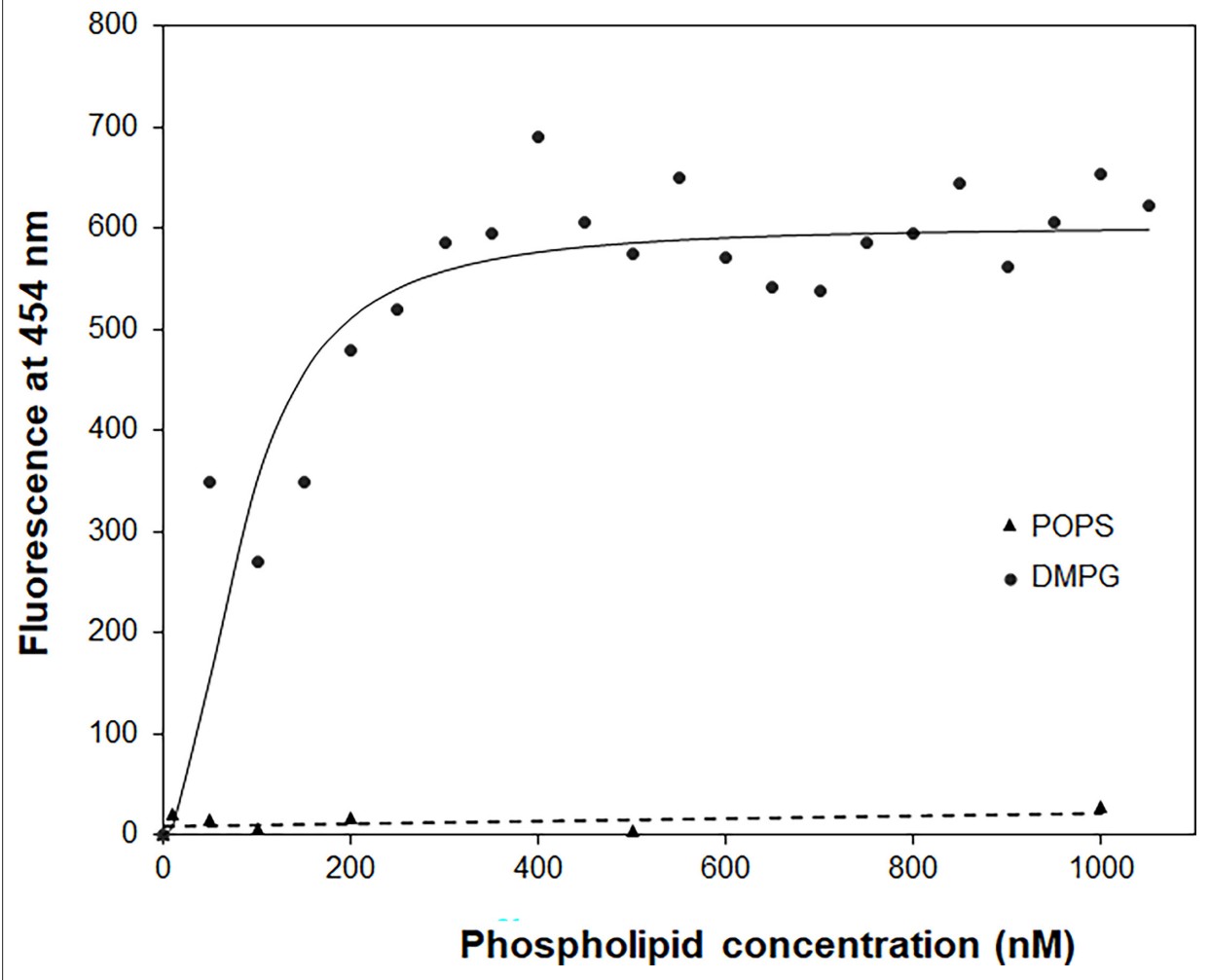

**Figure 4.** The binding affinity of daptomycin (Dap) for 1,2-dimyristoyl-*sn*-glycero-3-phosphorylglycerol (DMPG) by titration. The titration with DMPG was performed in 20 mM HEPES buffer (pH 7.57) containing 15 nM Dap and 1.67 mM CaCl₂. The solid line is the fitting curve using the equation $F=F_{max} \times [DMPG]^2/(K_D + [DMPG]^2)$, where F=fluorescence at 454 nm, which is derived from a binding model in which a fluorescent {Dap·2DMPG} complex is dissociated into nonfluorescent Dap and two DMPG molecules under the condition [DMPG] >> [Dap]. The dashed line is the linear trend line for the control titration of 1.5 µM Dap with 1-palmitoyl-2-oleoyl-*sn*-glycero-3-phospho-L-serine (POPS) in the same buffer containing 1.67 mM CaCl₂.

The online version of this article includes the following source data and figure supplement(s) for figure 4:

**Source data 1.** Kyn fluorescence change in titration with sub-micromolar DMPG.

**Figure supplement 1.** Determination of the critical micelle concentration (CMC) of 1,2-dimyristoyl-*sn*-glycero-3-phosphorylglycerol (DMPG).

**Figure supplement 1—source data 1.** The change of relative pyrene fluorescence intensity at 373 and 384 nm in the solution of DMPG at an increasing concentration from 7.8 nanomolar to 16 micromolar.

**Figure supplement 2.** Formation of the daptomycin (Dap)-2PG-Ca²⁺ complex.

reminiscent of the Dap-DMPG complex was found as suspension in aqueous phase. After being dissolved in DMSO, this precipitate was found to contain a pure Dap-containing complex with a retention time different from the Dap-DMPG complex in an HPLC analysis (*Figure 5A*). In addition, it was fluorescent with a spectrum characteristic of membrane-bound Dap like the Dap-DMPG complex (*Figure 5B*), strongly suggesting that the isolated precipitate was a similar phospholipid complex of Dap. In support of this, Dap was detected with a molecular ion at *m/z*=1642.66 ([M+Na]⁺, calcd. for C₁₇H₁₀₁N₁₇O₂₀Na: 1642.70) for both the cell-generated Dap-lipid complex and the Dap-DMPG complex in a comparative MALDI-ToF mass spectrometric analysis (*Figure 5C*). However, the phospholipid was not detected by MALDI-ToF mass spectrometry. High-resolution electron spray ionization mass spectrometry (HRESIMS) in negative-ion mode turned out to be more sensitive for negative

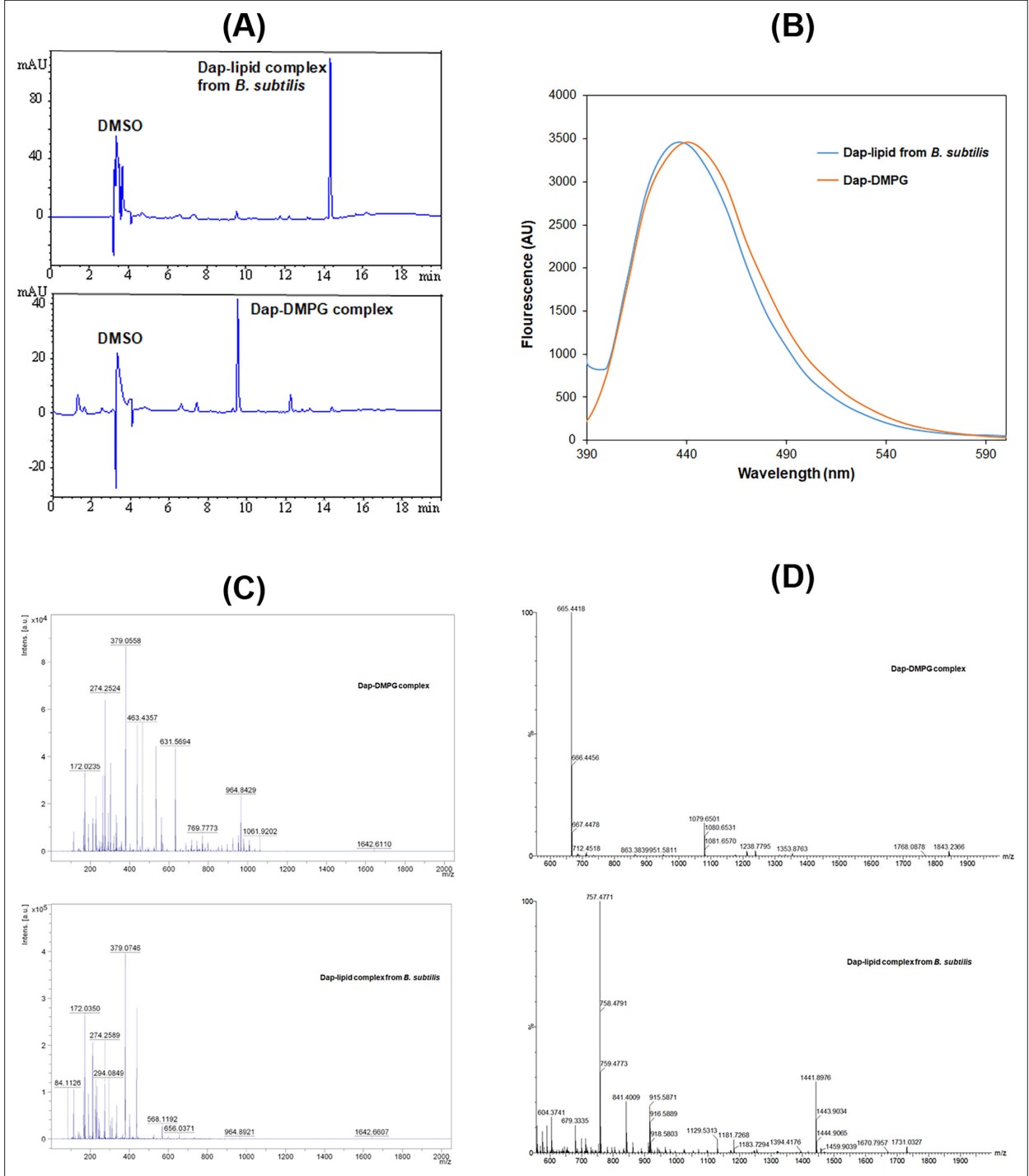

**Figure 5.** Stable daptomycin (Dap)-phosphatidylglycerol (PG) complexes formed in vitro and in *B. subtilis*. (**A**) Reverse-phase high-performance liquid chromatography (HPLC) chromatogram of the complexes. Samples were dissolved in 10% dimethyl sulfoxide (DMSO) and 90% water and filtered before injection into a Phenomenex Luna 5 μm C18 column (150×4.6 mm²). The complexes were eluted by water containing 0.1% trifluoroacetic acid and a linear gradient of acetonitrile from 5% to 95% over 20 min and detected by ultraviolet absorbance at 254 nm. (**B**) Fluorescence spectra of the two complexes with excitation at 365 nm. The complexes were dissolved in DMSO, and the fluorescence of the Dap-1,2-dimyristoyl-*sn*-glycero-3-phosphorylglycerol (DMPG) complex was scaled for easy comparison. (**C**) Comparative positive-ion MALDI-ToF mass spectrometry of the two complexes. All the peaks with *m/z*<500 were from the matrix after comparison with a control. (**D**) Comparative negative-ion high-resolution electron spray ionization mass spectrometry (ESIMS) analysis of the two complexes.

The online version of this article includes the following source data for figure 5:

**Source data 1.** The emission spectra of Dap in its lipid complex prepared in vitro or isolated from Bacillus subtilis.

phospholipids, affording DMPG molecular ion at 665.4418 ([M-H]$^-$, calcd. for $C_{34}H_{66}O_{10}P$: 665.4394) for the Dap-DMPG complex (*Figure 5D*). For the cell-generated complex, the phospholipid PG (32:0) was deduced from the main peak at *m/z*=757.4771 ([M+Cl]$^-$, calcd. for $C_{38}H_{75}O_{10}PCl$: 757.4792) in the negative-ion HRESIMS as an adduct with the matrix chloride ion, which could readily be formed in electron-spray ionization (*Wang et al., 2015*). PG (32:0) likely contains acyl chains derived from the two most abundant *iso*-15:0 and *anteiso*-17:0 branched fatty acids in *B. subtilis*, (*Nickels et al., 2017*) and its presence in the isolated complex as the major lipid species provides strong support for the specific interaction between Dap and PG. Taken together, all these results consistently indicated that Dap indeed forms a stable complex with PG in bacterial cells, which closely resembles the in vitro Dap-DMPG complex.

## Discussion

Dap is well known to target bacterial membrane for its bactericidal effect, but the mechanism of its membrane uptake is poorly understood. By monitoring the kinetics of membrane-induced fluorescence, here, we show that the calcium-dependent membrane uptake of Dap is divided into two stages. The first stage is reversible binding to the membrane surface, likely in the headgroup region, which happens very fast and reaches an equilibrium in milliseconds (*Figures 2 and 3*). The Dap binding capacity is dependent on the phospholipid, which is very low for neutral lipids PC and PE but is much higher for negative lipids, particularly CL (*Figure 2*). After this reversible surface binding, Dap is slowly inserted into the membrane in minutes (*Figures 2A and 3*) only in the presence of PG, culminating in the formation of a multicomponent complex. This complex contains calcium and two PG molecules with nanomolar affinity (*Figure 4*) and is easily formed in vitro and readily isolated from drug-treated bacterial cells (*Figure 5*). This high-affinity interaction provides an impeccable rationale for the neutralization of Dap by the PG-rich pulmonary surfactants (*Silverman et al., 2005*; *Pertel et al., 2008*). Moreover, the complete dependence of the membrane insertion on PG also explains why Dap selectively attacks Gram-positive bacteria without affecting mammalian cells, because PG is a major phospholipid in bacterial membrane but is a minor component in mammalian membrane.

Using CD spectroscopy, Dap was structurally examined in every step of its interaction with the membrane. In bulk aqueous solution, it takes a conformation with a characteristic positive peak at 230 nm and undergoes a large conformational change when bound to the headgroup region of membrane without PG, as evidenced from the dramatic change in CD (*Figure 3B*) that is consistent with previous investigations (*Jung et al., 2004*; *Lee et al., 2017*). It may not bind Ca$^{2+}$ in the bulk solution but requires the ion to bind the phospholipid headgroups, due to the observed release of the bound drug to the bulk solution with concomitant conformational reversal in the Ca$^{2+}$ sequestration experiments (*Figure 3*). In this surface-bound state, Dap is unable to insert into the acyl layer in the membrane, suggesting that its decyl tail is folded back into the hydrophobic part of the cyclodepsipeptide ring. In addition, due to the specificity for negative phospholipids (*Figure 2B and C*), this reversible binding of Dap likely involves both a nonspecific Ca$^{2+}$-mediated ionic interaction and a specific interaction with the remaining part of the headgroups.

In the irreversible insertion into the membrane, Dap undergoes a subtle conformational change from its surface-bound state to the inserted state, as seen from the small difference in their CD spectra (*Figure 3B*). When Ca$^{2+}$ is sequestrated, Dap undergoes another major conformational change from its inserted state to a new structure with CD features distinct from all other structures, supporting that the drug is still bound to Ca$^{2+}$ in its membrane-inserted state and that its decyl tail is fully extended and firmly embedded in the hydrophobic acyl layer of membrane. Besides binding Ca$^{2+}$, this inserted Dap should form a complex with PG in a 1:2 ratio as indicated by titration of Dap with DMPG in both micromolar (*Figure 2C*) and nanomolar range (*Figure 4*), which was formed easily in vitro and readily isolated from drug-treated *B. subtilis* (*Figure 5*). This complex is structurally similar to the tripartite Dap-PG-undecaprenyl lipid complex proposed to be responsible for the bactericidal effect of the drug (*Grein et al., 2020*). At present, it is not clear whether the Dap-Ca$^{2+}$-2PG complex contains one or more calcium ions (*Ho et al., 2008*; *Taylor et al., 2016*). For simplicity of discussion, only one calcium ion is assumed to be involved in the binding interaction to result in a quaternary Dap-Ca$^{2+}$-2PG complex.

Taking all the structural changes into consideration, we propose a mechanism for the Ca$^{2+}$-dependent uptake of Dap into the bacterial membrane, as shown in *Figure 6*. In the reversible binding,

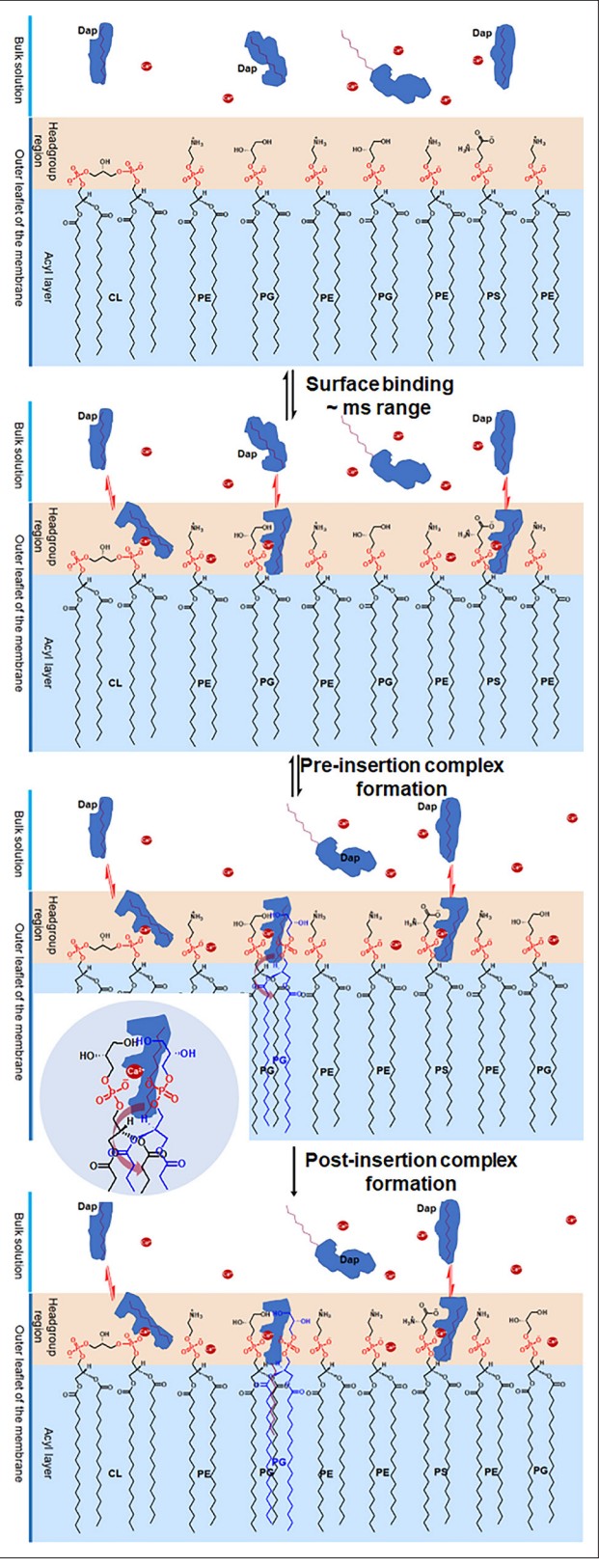

**Figure 6.** Proposed mechanism for the two-phased uptake of daptomycin (Dap) into bacterial membrane. In the first phase, Dap reversibly binds to negative phospholipids with a hidden tail in the headgroup region, where it combines with two phosphatidylglycerol (PG) molecules to form a pre-insertion complex. In the second phase, the hidden tail unfolds and irreversibly inserts into the membrane. The inset shows the headgroup of the pre-insertion

*Figure 6 continued on next page*

*Figure 6 continued*

complex with the broad arrow, showing the direction for the unfolding of the hidden tail. The red dots denote
$Ca^{2+}$.

Dap with a relatively random conformation is quickly bound to the headgroups of phospholipids likely through $Ca^{2+}$-dependent electrostatic interaction, accompanied by a large conformational change. With a hidden decyl tail, the surface-bound Dap is unable to insert into the membrane, even for the drug molecule bound by the PG headgroup. To enable membrane insertion, we propose that the bound Dap has to form a quaternary complex resembling the post-insertion Dap-$Ca^{2+}$-2PG complex, which is different from the latter in having a hidden rather than a fully extended decyl tail and is less stable. Subsequently, this pre-insertion complex undergoes structural change to expose the decyl tail of Dap and allows it to interact with the acyl chains of the two PG molecules to finish the insertion process, driven by the free energy to form the more stable post-insertion complex.

From this uptake mechanism, we can now understand how Dap recognizes bacteria and selectively accumulates in their membrane in a clinic setting. After absorption into the circulation, Dap is preferentially bound to the headgroup region of bacterial membrane due to the high content of PG and CL (*Nguyen et al., 2022*), very quickly in milliseconds, although it is also bound to mammalian cell surface in a smaller amount per cell due to the very low content or absence of PG or CL in plasma membrane (*Cockcroft, 2021*), but in substantial total quantity due to the overwhelmingly larger amount of host cells. However, Dap absorbed on the bacterial surface is continuously inserted into the acyl layer via the formation of a complex with PG in a timescale of minutes, whereas little irreversible insertion of Dap occurs on mammalian membrane due to the low content of PG while the bound Dap is continuously released to the circulation as the drug is depleted by the bacteria. The proposed requirement of the pre-insertion quaternary complex increases the threshold of PG content for the membrane insertion to happen and thus makes it less likely on the surface of mammalian cells that contain PG at a low level in the membrane. Consequently, most circulating drug is selectively inserted and accumulated in bacterial membrane within a short period of time where it exerts bactericidal effect via a mechanism still in debate (*Allen et al., 1987*; *Mengin-Lecreulx et al., 1990*; *Silverman et al., 2003*; *Zhang et al., 2014*; *Müller et al., 2016*; *Grein et al., 2020*).

In the Dap uptake, PG plays the dual role of recruiting the drug to the bacterial cell surface and enabling the irreversible insertion of the drug, of which both processes are critically affected in velocity and quantity by its membrane content. This PG dependence not only provides the basis for the reliance of Dap susceptibility on PG (*Hachmann et al., 2011*), but also is fully consistent with the finding that a decrease in membrane PG content inevitably leads to resistance to Dap, no matter whether the decrease is due to the gain-of-function mutations of MprF in the domain responsible for lysinylation of PG (*Friedman et al., 2006*; *Ernst et al., 2009*; *Yang et al., 2013*; *Mishra and Bayer, 2013*) or loss-of-function mutations of the PG synthase PgsA (*Hachmann et al., 2011*; *Hines et al., 2017*), or gain-of-function mutations of the cardiolipin synthase (Cls) to increase conversion of PG to CL (*Palmer et al., 2011*; *Arias et al., 2011*; *Davlieva et al., 2013*; *Jiang et al., 2019*). Understandably, when the PG biosynthesis is abolished due to complete inactivation of the PgsA activity, the Dap susceptibility is completely lost, resulting in a very high level of resistance (*Hines et al., 2017*; *Goldner et al., 2018*). However, a decreased PG content is found in only a fraction of resistant *Enterococcus* strains containing mutations in the LiaFSR operon and the PG metabolic enzymes GpdD and Cls in clinical isolates of *Enterococci* (*Arias et al., 2011*; *Tran et al., 2013*), suggesting that there exist PG-independent mechanisms to cause Dap resistance. In this connection, it is noteworthy that the PG content also appears not to be involved in the resistance caused by upregulated expression of *dltABCD* operon that increases the production of teichoic acids and their D-alanylation (*Bertsche et al., 2011*; *Yang et al., 2009*).

In summary, we have shown that Dap is first reversibly bound to the membrane surface and then irreversibly inserted into the acyl layer by forming a stable complex with PG. Interestingly, the stable complex may exchange one PG molecule with an undecaprenyl lipid to form the previously proposed bactericidal Dap-PG-undecaprenyl lipid tripartite complex to play a critical role in the antibacterial activity of the antibiotic. In addition, this stable complex also forms the basis of a unique mechanism for the bacterial uptake of Dap, in which the negative phospholipid PG plays a pivotal role to affect both the susceptibility and resistance to Dap. Nevertheless, it remains unknown how

the structural components of the drug affect nonspecific binding to the surface headgroup layer and specific binding with PG to influence the speed and dose of drug uptake. Elucidation of these structure-activity relationships may enable rational design of Dap analogs to counter the increasing drug resistance.

## Materials and methods

### Materials

Dap was purchased from MedChem Express. DMPC, CL from the bovine heart, FPP, calcium chloride, and calcium acetate were purchased from Sigma. The putative headgroup G3P was also purchased from Sigma. Tris base was purchased from Fisher Bioreagents. DMPG was purchased from Abcam. POPS and POPA were purchased from Avanti Polar Lipids. All lipids or chemicals were used directly without further treatment.

### Kinetics of the Kyn fluorescence change in micelles

Lipid stocks were prepared in 20 mM HEPES buffer with the pH adjusted to 7.57 at a concentration of 1.0 mM for DMPC and 0.50 mM for other lipids, including DMPG, POPA, POPE, POPS, CL, and FPP. In a typical kinetic measurement, a lipid micelle solution was prepared by mixing DMPC with or without another lipid in 20 mM HEPES (pH 7.57) and was sonicated for at least 4.5 hr (Ultrasonic Cleaner, 37 kHz, 100%). After the micelle solution was added with calcium chloride and Dap at the last, the kinetics of the Kyn fluorescence at 450 nm was immediately monitored in an Edinburgh FS-5 spectrofluorometer for 600 s with a time interval of 0.1 s and an excitation wavelength of 365 nm. The total volume of the solution was 300 μL (of which only 200 μL was added to the cuvette for reading) containing 167 μM DMPC with or without another lipid at a varied concentration, 1.67 mM $CaCl_2$, and 15.0 μM Dap in 20 mM HEPES (pH 7.57). After the kinetic measurement, an emission spectrum was recorded from 380 to 700 nm.

To study the effects of putative lipid headgroups on the fluorescence kinetics, their stocks were prepared at 50 mM with the pH adjusted to 7.57. The putative headgroup was then added at a preset concentration to the micelle solution for the kinetic measurements. In these measurements, final DMPC concentration was set at 80 μM, and final DMPG concentration was set at 10 μM (or 11.5%), while all other components and the instrumental parameters remained unchanged.

In the calcium sequestration experiments, an EGTA stock was prepared at 42.85 mM and dissolved in HEPES by NaOH and adjusted to pH 7.57 by HCl. It was then added at a saturating concentration (10 mM) to the micelle solution at the end of the 600 s kinetic measurement, and the fluorescence kinetics was monitored at 450 nm for another 10 min or a longer time (up to 70 min). The micelle solution contained 167 μM DMPC with or without another phospholipid, namely DMPG, or CL, or POPS at 20.8 μM (or 11.5%). All other conditions were the same as in other kinetic measurements. Similar experiments were also performed with a different combination of phospholipids such as 167 μM DMPC and 10.4 μM DMPG.

### Kinetics of the Kyn fluorescence change in vesicles

Phospholipid vesicles were prepared in a varied ratio from a DMPC stock and the stock of another phospholipid, namely DMPG, CL, or POPS, of which both were in chloroform at a concentration of 500 and 125 μM, respectively. After mixing, the solvent chloroform was evaporated by a Rotavap under reduced pressure at 37°C for 15 min, to form concentric rings at the bottom of the round bottom flask. The phospholipids were further dried under vacuum for 2 hr and then hydrated overnight in 20 mM HEPES (pH 7.57). The hydrated phospholipid film was subjected to five freeze-thaw cycles, in each of which the vesicles were frozen in liquid nitrogen for 1 min, followed by 5 min thawing and 10 s vortex. Finally, the vesicle suspension in a 1 mL syringe was pushed through a 100 nm filter in an Avanti polar extruder with filter supports. Vesicles went through at least 10 passes, and the extruder was warmed on a hot plate to ease the passing of vesicles. The vesicle suspension was diluted to the preset concentration of the phospholipid and used to interact with Dap for kinetic monitoring of the Kyn fluorescence similar to the measurements using the micelles.

## Dap conformational change by CD spectroscopy

CD was used to characterize the structure of Dap in its interaction with calcium, G3P, and lipid micelles under various conditions using a Chirascan Circular Dichroism Spectrometer (Applied Photophysics). The samples were prepared to contain 0.30 mg/mL Dap from stocks adjusted to pH 7.57, to avoid pH change after mixing of different components. To reduce background, 20 mM Tris buffer (pH 7.57) was used instead of the HEPES buffer, and calcium acetate was used at 1.0 mM to replace calcium chloride used in the fluorescence experiments. Each sample was scanned 10 times in a 1 mm Hellma Quartz cuvette from 180 to 270 nm with 1 nm bandwidth, 1 nm step size, and 0.5 s per point (approximately 73 s). The CD spectrum was then averaged from the 10 scans using the Chirascan software after auto-subtraction with the background signal of a blank buffer.

For Dap interaction with model membranes, the micelle solution was prepared from lipid stocks adjusted to pH 7.57 to contain one or two lipids at preset concentrations, sonicated for 4 hr, and then added 1.0 mM calcium acetate and 0.30 mg/mL Dap for incubation at room temperature for at least 15 min before the CD spectra were taken. To maximize the drug reversibly bound to the micelle surface, pure CL micelle solution was used at 720 μM (higher concentration led to precipitation) to bind Dap in the presence of 1 mM calcium acetate for CD spectroscopy. Similarly, pure DMPG micelles at 720 μM were used in the presence of 1 mM calcium acetate to ensure complete membrane insertion of Dap for structural characterization by CD spectroscopy. To characterize the conformation of membrane-bound Dap after calcium removal, EGTA in 20 mM HEPES at pH 7.57 was added at a saturating concentration of 10 mM to the solution of Dap bound to micelles of either pure CL or pure DMPG at 720 μM, incubated at room temperature for at least 30 min and then subjected to CD spectroscopy.

## Interaction of Dap with G3P by NMR spectroscopy

For the NMR analysis, stocks of calcium chloride, G3P, and Dap were prepared in 20 mM HEPES with 10% $D_2O$ (vol/vol) and pH/pD adjusted to 5.40. They were used to prepare the samples (in 1000 μL) that contained 1.0 mM Dap, 1 mM $CaCl_2$, and G3P at a varied concentration at 0 mM, 1.0 mM, 4.0 mM, 8.0 mM, 12.0 mM, 16.0 mM, and 20.0 mM in the same deuterated buffer. Controls containing 1.0 mM Dap or 20 mM G3P were also prepared for the NMR experiments in the same buffer. The 1D $^1$H-NMR spectrum and standard 2D spectra including TOCSY (spin lock system was 75 mS), COSY, and HMBC were recorded for each sample on an 800 MHz Varian spectrometer at room temperature (23°C). All spectra were analyzed using Mestrenova, and the cross-peaks in the fingerprint region were assigned by comparing the $N_\alpha H$-$H_\alpha$-$H_\beta$-$H_\gamma$ correlation patterns of the amino acid residues of Dap in TOCSY with those obtained for the drug at pH 5.0 in a previous study (*Ball et al., 2004*). The assignments are shown in *Supplementary file 1*, and the TOCSY comparison is shown in *Figure 3—figure supplement 2*.

## CMC of DMPG and titration of Dap with DMPG in the nanomolar range

The CMC of DMPG was determined using a reported method (*Goddard et al., 1985*). Briefly, pyrene (~20 mg) was crushed to powder and added to 5.0 mL to obtain a 1.0 mM stock of DMPG in 20 mM HEPES buffer (pH 7.57). The mixture was sonicated for 4.5 hr, filtered to remove undissolved pyrene, and then diluted in the appropriate medium (either pure water or 20 mM HEPES, pH 7.57) to contain DMPG at a varied concentration from 10 nM to 10 μM. All samples were added calcium chloride to a final concentration of 1.67 mM, and their emission spectra from 340 to 600 nm were recorded on an FS-5 Edinburgh spectrofluorometer with an excitation wavelength at 330 nm and a bandwidth of 3 nm. The ratio of the fluorescence intensity at 473 and 484 nm was plotted against the DMPG concentration to determine CMC, as shown in *Figure 3—figure supplement 2*.

For the fluorescence titration at low concentrations, each sample was prepared in 300 μL to contain 1.67 mM $CaCl_2$, 15 nM Dap, and DMPG at a varied concentration ranging from 15 nM to 10 μM in an appropriate medium (pure water or 20 mM HEPES, pH 7.57). DMPG was added from a stock of DMPG at 5.0 μM in 20 mM HEPES (pH 7.57), while Dap was added from a stock at 150 nM. These samples were then recorded for their emission spectra in the range of 380–700 nm on an FS-5 Edinburgh spectrofluorometer with the excitation wavelength set at 365 nm. The bandwidth was set at 3 nm for both the excitation and emission. The emission value at 450 nm was then plotted against the DMPG

concentration (as shown in *Figure 4A*). A control titration was performed similarly using 1.5 μM Dap against POPS in the concentration range of 0–1.0 μM.

## Isolation of Dap-lipid complex from drug-treated *B. subtilis*

*B. subtilis* 168 was grown in 500 mL LB for 12 hr at 37°C inoculated with 1 mL overnight LB culture from a single colony, harvested, resuspended in 10 mL fresh LB, and then added 0.375 mg/mL Dap and 0.75 mg/mL $CaCl_2$ for incubation at 37°C for another 2 hr. The drug-treated cells were pelleted, washed once with 10 mL 0.9% NaCl solution, resuspended in 10 mL 1 mM NaCl, and lysed by sonication (1 s sonication, 2 s pause, Instrument: JY92-II) for 1 hr over ice. The lysate was centrifuged to remove the cell debris, and the supernatant was ultracentrifuged at 125,000×$g$ at 4°C for 1 hr in Hitachi CP80WX Preparative Ultracentrifuge. After removing the supernatant, the pellet was suspended in 2 mL of 0.9% NaCl solution and extracted with 2 mL chloroform to observe a layer of white precipitate at the phase interface. At this point, 1 mL of 0.1 M SDS solution was added to the aqueous phase, and the mixture was vortexed for 1 min. 3 mL of DMSO was then added to quench the bubbles, and the aqueous phase appeared cloudy. The aqueous layer was separated and centrifuged at high speed to obtain a small amount of white pellet, which exhibits fluorescent emission typical of membrane-absorbed Dap at 365 nm excitation after suspension in water or being dissolved in DMSO. In a parallel control experiment starting from the same amount of cell culture treated with $CaCl_2$ but without Dap, the aqueous phase was clear without any precipitate after addition of SDS, indicating that SDS dissolved all protein precipitates formed from the chloroform extraction. Like the Dap-DMPG complex, the final precipitate or pellet obtained from the drug-treated cells was soluble in DMSO and partly soluble in $CH_2Cl_2$ and was subject to analysis by reverse-phase HPLC, MALDI-ToF mass spectrometry, and high-resolution ESIMS.

## Acknowledgements

This work was supported by GRF16102121 from the Research Grants Council of Hong Kong SAR and the grant 21877094 from the National Natural Science Foundation of China.

## Additional information

### Funding

| Funder | Grant reference number | Author |
| --- | --- | --- |
| Research Grants Council, University Grants Committee | GRF16102121 | Zhihong Guo |
| National Natural Science Foundation of China | 21877094 | Zhihong Guo |

The funders had no role in study design, data collection and interpretation, or the decision to submit the work for publication.

### Author contributions

Pragyansree Machhua, Data curation, Formal analysis, Investigation, Methodology, Writing – review and editing; Vignesh Gopalakrishnan Unnithan, Formal analysis, Investigation, Methodology; Yu Liu, Yiping Jiang, Lingfeng Zhang, Investigation, Methodology, Writing – review and editing; Zhihong Guo, Conceptualization, Formal analysis, Supervision, Funding acquisition, Writing - original draft, Project administration, Writing – review and editing

### Author ORCIDs

Pragyansree Machhua https://orcid.org/0000-0001-9425-130X
Yu Liu https://orcid.org/0000-0001-6179-7699
Lingfeng Zhang https://orcid.org/0009-0009-5282-8022
Zhihong Guo https://orcid.org/0000-0003-0374-8412

Reviewer #3 (Public review): https://doi.org/10.7554/eLife.93267.3.sa1
Author response https://doi.org/10.7554/eLife.93267.3.sa2

## Additional files

### Supplementary files
Supplementary file 1. Assignment of the COSY signals of daptomycin (Dap) in the fingerprint region.
MDAR checklist

### Data availability
All data generated or analysed during this study are included in the manuscript and supporting files.

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
