## [Editor Report · eLife Assessment]

This **valuable** study describes the molecular mechanism of daptomycin insertion into bacterial membranes. The authors provide **solid** in vitro evidence for the early events of daptomycin interaction with phospholipid headgroups and stronger, specific interaction with phosphatidylglycerol. This work will be of interest to bacterial membrane biologists and biochemists working in the antimicrobial resistance field.

---

## [Referee Report · Reviewer #3 (Public review)]

Summary:

Machhua et al. in their work focused on unravelling the molecular mechanism of daptomycin binding and interaction with bacterial cell membranes. Daptomycin (Dap) is an acidic, cyclic lipopeptide composed of 13 amino acids, known for preferential binding to anionic lipids, particularly phosphatidylglycerol (PG), which are prevalent components in the membranes of Gram-positive bacteria. The process of binding and antimicrobial efficacy of Dap are significantly influenced by the ionic composition of the surrounding environment, especially the presence of Ca2+ ions. The authors underscore the presence of significant knowledge gaps in our understanding of daptomycin's mode of action. Several critical questions remain unanswered, including the basis for selective recognition and accumulation in membranes of Gram-positive strains, the specific role of Ca2+ ions in this process, and the mechanisms by which daptomycin binds to and inserts into the cell membrane.

Dap is intrinsically fluorescent due to its kynurenine residue (Kyn-13) and this property allows direct imaging of Dap binding to model cell membranes without the need of additional labeling. Taking advantage of this Dap autofluorescence, authors monitored the emission intensity of micelles, composed of varying DMPG content upon their exposure to Dap and compared it with the kinetics of fluorescence observed for zwitterionic DMPC and other negatively charged lipids such as cardiolipin (CA), POPA and POPS. The authors noted that the linear relationship between DMPG content and Dap fluorescence is strongly lipid-specific, as it was not observed for other anionic lipids. The manuscript sheds light on the specificity of Dap's interaction with CA and DMPG lipids. Through Ca2+ sequestration with EGTA, the authors demonstrated that the binding of Dap with CA is reversible, while its interaction with DMPG results in the irreversible insertion of Dap into the lipid membrane structure, caused by the significant conformational change of this lipopeptide. The formation of a stable DMPG-Dap complex was also verified in bacterial cells isolated from Gram-positive bacteria *B. subtilis*, where Dap exhibited a permanent binding to PG lipids.

Altogether, the authors endeavored to illuminate novel insights into the molecular basis of Dap binding, interaction, and the mechanism of insertion into bacterial cell membranes. Such understanding holds promise for the development of innovative strategies in combating drug resistance and the emerging of the so-called superbugs.

Strengths:

- The manuscript by Machhua et al. provides a comprehensive analysis of the Dap mechanism of binding and interaction with the membrane. It discusses various aspects of this, only apparently trivial interaction such as the importance of PG presence in the membrane, the impact of Ca2+ ions, and different mechanisms of Dap binding with other negatively charged lipids.

- The authors focused not only on model membranes (micelles) but also extended their research to bacterial cell membranes obtained from *B. subtilis*

- The research is not only a report of the experimental findings but tries to give potential hypotheses explaining the molecular mechanisms behind the observed results

Weaknesses:

- The authors overestimate their findings, stating that they propose a novel mechanism of Dap interaction with bacterial cell membranes. This research is the extension of the hypotheses that have already been reported.

- The literature study and overall discussion about the mechanism of action of Ca2+ ions or conformational changes of daptomycin could be improved.

---

## [Author Response]

The following is the authors’ response to the original reviews.

**Reviewer #1 (Recommendations For The Authors):**
Summary:In this manuscript, the molecular mechanism of interaction of daptomycin (DAP) with bacterial membrane phospholipids has been explored by fluorescence and CD spectroscopy, mass spectrometry, and RP-HPLC. The mechanism of binding was found to be a two-step process. A fast reversible step of binding to the surface and a slow irreversible step of membrane insertion. Fluorescence-based titrations were performed and analysed to infer that daptomycin bound simultaneously two molecules of PG with nanomolar affinity in the presence of calcium. Conformational change but not membrane insertion was observed for DAP in the presence of cardiolipin and calcium.Strengths:The strength of the study is skillful execution of biophysical experiments, especially stoppedflow kinetics that capture the first surface binding event, and careful delineation of the stoichiometry.Weaknesses:The weakness of the study is that it does not add substantially to the previously known information and fails to provide additional molecular details. The current study provides incremental information on DAP-PG-calcium association but fails to capture the complex in mass spectrometry. The ITC and NMR studies with G3P are inconclusive. There are no structural models presented. Another aspect missing from the study is the reconciliation between PG in the monomer, micellar, and membrane forms.

Besides the two-stage process, another important finding in the current work is the stable complex that plays a critical role in the drug uptake both in vitro and in *B. subtilis*. This complex has been shown to be a stable species in HPLC and its binding stoichiometry and affinity have been quantitatively characterized. The complex may not be stable enough in gas phase to be detected in the MS analysis, which was designed to detect the phospholipid and Dap components, not the complex itself. The structural model of this complex is clearly proposed and presented in Figure 6.

The NMR and ITC studies have a very clear conclusion that Dap has a weak interaction with the PG headgroup alone, which is unable to account for the Dap-PG interaction observed in the fluorescence studies. Thus, the whole PG molecule has to be involved in the interaction, leading to the discovery of the stable complex.

**Reviewer #2 (Recommendations For The Authors):**
(1) I appreciate and agree with the comment that there are stages of daptomycin insertion, and these might involve the formation of different complexes with different binding partners (e.g. pre-insertion vs quaternary vs bactericidal). However, it seems like lipid II is an apparent participant in daptomycin membrane dynamics (Grein et al. Nature Communications 2020). It's not clear why this was excluded from analysis by the authors, or what basis there is for the discussion statement that the quaternary complex can shift into the bactericidal complex by exchanging 1 PG for lipid II.

We agree that lipid II and other isoprenyl lipids may be involved in the uptake and insertion of daptomycin into membrane according to the results of the *Nat. Comm.* paper. However, these isoprenyl lipids are very small components of the membrane in comparison to PG and their contribution to the drug uptake is thus expected to be much less significant. Nonetheless, we included farnesyl pyrophosphate (FPP) as an analog of bactoprenol pyrophosphate (C55PP), which was reported to have the same promoting effect as lipid II in the previous study, in our study but found no promoting effect in the fluorescence assay (Fig. 2B). In addition, no complex was formed when FPP replaced PG in our preparation and analysis of the drug-lipid complex. In consideration of these negative results and the expected small contribution, other isoprenyl lipids or their analogs were not included in the study.

The statement of forming the proposed bactericidal complex from the identified complex is a speculation that is possible only when lipid II has a higher affinity for Dap than a PG ligand. To avoid confusion, we deleted the sentence’ in the revision.

(2) The detailed examination of daptomycin dynamics, particularly on the millisecond scale, in this paper is ideal for characterizing the effect of lipid II on daptomycin insertion. It would be helpful to either include lipid II in some analyses (micelle binding, fluorescence shifts, CD) or at least address why it was excluded from the scope of this work.

As mentioned in the response to the first comment, we did not exclude isoprenyl lipids in our study but used some of their analogs in the fluorescence assay. Besides FPP mentioned above, we also tested geranyl pyrophosphate and geranyl monophosphate but obtained the same negative results. Lipid II was not directly used because it is one of the three isoprenyl lipids reported to have the same promoting effects in the *Nat. Comm*. paper and also because its preparation is not easy. Even if lipid II were different from other isoprenyl lipids in promoting membrane binding, its contribution is likely negligible at the reversible stage compared to the phospholipids because of its minuscule content in bacterial membrane. This is the main reason we did not use the isoprenyl lipids in the fast kinetic study (this stage only involves reversible binding, not insertion).

(3) Grein et al. 2020 saw that PG did not have a strong effect on daptomycin interaction with membranes. I believe this discrepancy is more likely due to the complex physical parameters of supported bilayers versus micelles/vesicles or some other methodological variable, but if the authors have more insight on this, it would be valuable commentary in the discussion.

We totally agree that the discrepancy is likely due to the different conditions in the assays. It is hard to tell exactly what causes the difference. Thus, we did not attempt to comment on the cause of this difference in the discussion.

(4) Isolation of the daptomycin complex from *B. subtilis* cells clearly had different traces from the in vitro complex; is it possible that lipid II is present in the *B. subtilis* complex? If not, a time-course extraction could be useful to support the model that different complexes have different activities. Isolates from early-stage incubation with daptomycin may lack lipid II but isolates from longer incubations may have lipid II present as the complex shifts from insertion to bactericidal.

From the day we isolated the complex from *B. subtilis*, we have been looking for evidence for the previously proposed lipid complexes containing lipid II or other isoprenyl lipids but have not been successful. We did not see any sign of lipid II or other isoprenyl lipids in the MALDI or ESI mass spectroscopic data. The minute peaks in the HPLC traces are not the expected complexes in separate LC-MS analysis. However, this does not mean that such complexes are not present in the isolated PG-containing complex because: (1) the amount of such complexes may be too small to be detected due to the low content of the isoprenyl lipids; (2) the isoprenyl lipids, particularly lipid II, are not easily ionizable due to their size and unique structure for detection in mass spectrometry.

We don’t think the drug treatment time is the reason for the failure in detecting lipid II or other isoprenyl lipids. In our reported experiment, the cells were treated with a very high dose of Dap for 2 hours before extraction. In a separate experiment done recently, we treated *B. subtilis* at 1/3 of the used dose under the same condition and found all treated cells were dead after 1 hour in a titration assay, consistent with the results from reported time-killing assays in the literature. From this result, the proposed bactericidal lipid-containing complex should have been formed in the treated cells used in our extraction and isolated along with the PG-containing complex. It was not detected likely due to the reasons discussed above. To avoid the interference of the PG-containing complex, a large amount of bacterial cells might have to be treated at a low dose to isolate enough amount of the lipid II-containing complex for identification. However, isolation or identification of the lipid II-containing complex is outside the scope of the current investigation and is therefore not pursued.

(5) Part of the daptomycin mechanism of interacting with bacterial membranes involves the flipping of daptomycin from one leaflet to another. There was some mentioned work on the consistency of results between micelles and vesicles, but the dynamics or existence of a flipping complex in the bilayer system wasn't addressed at all in this paper.

The current investigation makes no attempt to solve all problems in the daptomycin mode of action and is limited to the uptake of the drug, up to the point when Dap is inserted into the membrane. Within this scope, flipping of the complex is not yet involved and is thus irrelevant to the study. How the complex is flipped and used to kill the bacteria is what should be investigated next.

(6) The authors mention data with phosphatidylethanolamine in the text, but I could not find the data in the main or supplemental figures. I recommend including it in at least one of the figures.

It is much appreciated that this error is identified. The POPE data was lost when the graphic (Fig. 2B) was assembled in Adobe to create Figure 2. We re-draw the graphic and reassemble the figure to solve this problem. Fig. 2B has also been modified to use micromolar for the concentration of the lipids.

(7) Readability point: I'd suggest some consistency in the concentrations mentioned. Making the concentrations either all molar-based or all percentage-based would make comparison across figures easier.

As suggested, we have changed the % into micromolar concentrations in Fig. 2B and also in Fig. 3A.

(8) The model figure is quite difficult to interpret, particularly the final stage of the tail unfolding. I recommend the authors use a zoomed-in inset for this stage, or at least simplify the diagram by removing the non-participating lipid structures. The figure legend for the model figure should also have a brief description of the events and what the arrows mean, particularly the POPS PG arrow in the final panel of the figure. I am assuming here the authors are implying that daptomycin can transiently interact with one lipid species and move to another, but the arrow here suggests that daptomycin is moving through the lipid headgroup space.

We really appreciate the suggestions. As suggested, we put an inset to show the preinsertion complex more clearly. In addition, we have removed the green arrows originally intended to show the re-organization/movement of the phospholipids. Moreover, the legend is changed to ‘Proposed mechanism for the two-phased uptake of Dap into bacterial membrane. In the first phase, Dap reversibly binds to negative phospholipids with a hidden tail in the headgroup region, where it combines with two PG molecules to form a pre-insertion complex. In the second phase, the hidden tail unfolds and irreversibly inserts into the membrane. The inset shows the headgroup of the pre-insertion complex with the broad arrow showing the direction for the unfolding of the hidden tail. The red dots denote Ca2+.’

(9) The authors listed the Kd for daptomycin and 2 PG as 7.2 x 10-15 M2. Is this correct? This is an affinity in the femtomolar range.

Please note that this Kd is for the simultaneous binding of two PG molecules, not for the binding of a single ligand that we usually refer to. Assuming that each PG contributes equally to this interaction, the binding affinity for each ligand is then the squared root of 7.2 x 10-15 M2, which equals to 8.5 x 10-8 M. This is equivalent to a nanomolar affinity for PG and is a reasonably high affinity.

**Reviewer #3 (Recommendations For The Authors):**
(1) The authors reported an increase in daptomycin intensity with the increasing amount of negatively charged DMPG. A similar observation has been reported for GUVs, however, the authors did not refer to this paper in their manuscript: E. Krok, M. Stephan, R. Dimova, L. Piatkowski, Tunable biomimetic bacterial membranes from binary and ternary lipid mixtures and their application in antimicrobial testing, Biochim. Biophys. Acta - Biomembr. 1865 (2023) [1]. This paper is also consistent with the authors' observation that there is negligible fluorescence detected for the membranes composed of PC lipids upon exposure to the Dap treatment.

As suggested, this paper is cited as ref. 29 in the revision by adding the following sentence at the end of the section ‘Dependence of Dap uptake on phosphatidylglycerol.’: ‘PG-dependent increase of the steady-state fluorescence was also observed in giant unilamellar vesicles (GUVs).29’. The numbering is changed accordingly for the remaining references.

(2) Please include the plot of the steady-state Kyn fluorescence vs the content of POPA (Figure 2C shows traces for DMPG, CL, and POPS). Both POPA and POPS lipids are negatively charged, however, POPS seems to interact with Dap, while POPA does not. In my opinion, this observation is really interesting and might deserve a more thorough discussion. The authors might want to describe what could be the mechanism behind this lipid-specific mode of binding.

As suggested, a plot is now added for POPA in Fig. 2C, which is basically a flat line without significant increase for the Kyn fluorescence. Indeed, the different effect of the negative phospholipids is very interesting, indicating that the reversible binding of Dap to the lipid surface is dependent not only on the Ca2+-mediated ionic interaction but also the structure of the headgroup. In other words, Dap recognizes the phospholipids at the surface binding stage. Considering this headgroup specificity, the last sentence in the second paragraph in “Discussion’ is changed from ‘In addition, due to the low lipid specificity, this reversible binding likely involves Ca2+-mediated ionic interaction between Dap and the phosphoryl moiety of the headgroups.’ to ‘In addition, due to the specificity for negative phospholipids (Fig. 2B and 2C), this reversible binding of Dap likely involves both a nonspecific Ca2+-mediated ionic interaction and a specific interaction with the remaining part of the headgroups.’

(3) The authors write that they propose a novel mechanism for the Ca2+-dependent insertion of Dap to the bacterial membrane, however, they rather ignored the already published findings and hypotheses regarding this process. In fact the role of Ca2+, as well as the proposed conformational changes of Dap, which allow its deeper insertion into the membrane are well known:The role of Ca2+ ions in the mechanism of binding is actually three-fold: (i) neutralization of daptomycin charge [2], (iii) creating the connection between lipids and daptomycin and (iii) inducing two daptomycin conformational changes. It should be noted that the interactions between calcium ions and daptomycin are 2-3 orders of magnitude stronger than between daptomycin and PG lipids [3,4]. Thus, upon the addition of CaCl2 to the solution, the divalent cations of calcium bind preferentially to the daptomycin, rather than to the negatively charged PG lipids, which results in the decrease of daptomycin net negative charge but also leads to its first conformational change [4]. Upon binding between calcium ions and two aspartate residues, the area of the hydrophobic surface increases, which allows the daptomycin to interact with the negatively charged membrane. In the next step, Ca2+ acts as a bridge connecting daptomycin with the anionic lipids. This event leads to the second conformational change, which enables deeper insertion of daptomycin into the lipid membrane and enables its fluorescence [4]. The overall mechanism has a sequential character, where the binding of daptomycin-Ca2+ complex to the negatively charged PG (or CA) occurs at the end.The authors should focus on emphasizing the novelty of their manuscript, keeping in mind the already published paper.

We agree with the comments on the three general roles of calcium ion in the Dap interaction with membrane. The current investigation does not ignore the previous findings, which involve many more works than mentioned above, but takes these findings as common knowledge. Actually, the role of calcium ion is not the focus of current work. Instead, the current work focuses on how the drug is taken up and inserted into the membrane in the presence of the ion and how its structure changes in this process. With the known roles of calcium ion in mind, we propose an uptake mechanism (Fig. 6) that shows no conflict with the common knowledge.

We would like to point out that the ‘deeper insertion into the membrane’ in the comment is different from the membrane insertion referred to in our manuscript. This ‘deeper insertion’ still remains in the reversible stage of binding to the membrane surface because all negative phospholipids can do this (causing a conformational change and fluorescence increase, as quantified in Fig.2C) but now we know that only PG can enable irreversible membrane insertion because of our work. In addition, the comment that calcium binding to daptomycin causes first conformational change is not supported by our finding that no conformational change is found for Dap in the presence of calcium in a lipid-free environment (Fig. 3B). One important aspect of novelty and contribution of our work is to clear up some of these ambiguities in the literature. Another contribution of our work is to demonstrate the formation of a stable complex between Dap and PG with a defined stoichiometry and its crucial role in the drug uptake.

(4) One paragraph in the section "Ca2+- dependent interaction between Dap and DMPG" is devoted to a discussion of the formation of precipitate upon extraction of DMPG-containing micelles, exposed to Dap in the calcium-rich environment. Contrary, in the absence of Dap, no precipitate was detected. The authors did not provide any visual proof for their statement. Please include proper photographs in the supplementary information.

The precipitate formed upon extraction of the DMPG-containing micelles was too little to be visually identifiable but could be collected by centrifugation and detected by fluorescence or HPLC after dissolving in DMSO. For visualization, we show below the precipitate formed using higher amount of Dap and DMPG. The Dap-DMPG-Ca2+ complex (left tube) was formed by mixing 1 mM Dap, 2 mM DMPG and 1 mM Ca2+ and the control (right tube) was a mixture of 2 mM DMPG and 1 mM Ca2+. This is now added as Fig. S7 in the supplementary information (the index is modified accordingly) and cited in the main text.

(5) The authors wrote that it is not clear how many calcium ions are bound to Dap-2PG complex (page 11, Discussion section). There are already reports discussing this issue. I recommend citing the paper discussing that exactly two Ca2+ ions bind to a single Dap molecule: R. Taylor, K. Butt, B. Scott, T. Zhang, J.K. Muraih, E. Mintzer, S. Taylor, M. Palmer, Two successive calcium-dependent transitions mediate membrane binding and oligomerization of daptomycin and the related antibiotic A54145, Biochim. Biophys. Acta - Biomembr. 1858, (2016) 1999-2005 [5]

We were aware of the cited work that shows binding of two Ca2+ but also noted that there are more works showing one Ca2+ in the binding, such as the paper in [Ho, S. W., Jung, D., Calhoun, J. R., Lear, J. D., Okon, M., Scott, W. R. P., Hancock, R. E. W., & Straus, S. K. (2008), Effect of divalent cations on the structure of the antibiotic daptomycin. *European Biophysics Journal*, *37*(4), 421–433.]. That was the reason we said ‘it is not clear how many calcium ions are bound to Dap-2PG complex’. Now, both papers are cited (as Ref. #33, 34) to support this statement.

(6) The authors wrote two contradictory statements:- PG cannot be found in mammalian cell membranes:"Moreover, the complete dependence of the membrane insertion on PG also explains why Dap selectively attacks Gram-positive bacteria without affecting mammalian cells, because PG is present only in bacterial membrane but not in mammalian membrane. " (Page 10, Discussion section, last sentence of the first paragraph)"However, Dap absorbed on bacterial surface is continuously inserted into the acyl layer via formation of complex with PG in a time scale of minutes, whereas no irreversible insertion of Dap occurs on mammalian membrane due to the absence of PG while the bound Dap is continuously released to the circulation as the drug is depleted by the bacteria." (Page 13, Discussion section)- PG in trace amounts is present in mammalian membranes:"The proposed requirement of the pre-insertion quaternary complex increases the threshold of PG content for the membrane insertion to happen and thus makes it impossible on the surface of mammalian cells even if their plasma membrane contains a trace amount of PG." (Page 13, Discussion section).In fact, phosphatidylglycerol comprises 1-2 mol% of the mammalian cell membranes. Please, correct this information, which in this form is misleading to the readers.

We appreciate the comments about the PG content in mammalian cells. Changes are made as listed below:

(1) p10, the sentence is changed to ‘Moreover, the complete dependence of the membrane insertion on PG also explains why Dap selectively attacks Gram-positive bacteria without affecting mammalian cells, because PG is a major phospholipid in bacterial membrane but is a minor component in mammalian membrane.’

(2) p13, the sentence is changed to ‘However, Dap absorbed on bacterial surface is continuously inserted into the acyl layer via formation of complex with PG in a time scale of minutes, whereas little irreversible insertion of Dap occurs on mammalian membrane due to the low content of PG while the bound Dap is continuously released to the circulation as the drug is depleted by the bacteria.’

(3) p13, another sentence is modified to ‘The proposed requirement of the pre-insertion quaternary complex increases the threshold of PG content for the membrane insertion to happen and thus makes it less likely on the surface of mammalian cells that contain PG at a low level in the membrane.’

(7) Please include information that Dap is effective only against Gram-positive bacteria and does not show antimicrobial properties against Gram-negative strains. The authors focused on emphasizing that Dap does not affect mammalian membranes, most likely due to the low PG content, however even membranes of Gram-negative bacteria are not susceptible to the Dap, despite the relatively high content of negatively charged PG in the inner membrane (e.g. inner cell membrane of *E. coli* has ~20% PG).

The requested information is already included in ‘Introduction’. In this part, Dap is introduced to be only active against Gram-positive bacteria, implicating that it is not active against Gram-negative bacteria. The reason Dap is inactive against *E. coli* or other Gramnegative bacteria is because the outer membrane prevents the antibiotic from accessing the PG in the inner membrane to cause any harm. When the outer membrane is removed, Dap will also attack the plasma membrane of Gram-negative bacteria.

Literature cited in the comments:

(1) E. Krok, M. Stephan, R. Dimova, L. Piatkowski, Tunable biomimetic bacterial membranes from binary and ternary lipid mixtures and their application in antimicrobial testing, Biochim. Biophys. Acta - Biomembr. 1865 (2023). https://doi.org/10.1101/2023.02.12.528174.

(2) S.W. Ho, D. Jung, J.R. Calhoun, J.D. Lear, M. Okon, W.R.P. Scott, R.E.W. Hancock, S.K. Straus, Effect of divalent cations on the structure of the antibiotic daptomycin, Eur. Biophys. J. 37 (2008) 421-433. https://doi.org/10.1007/S00249-007-0227-2/METRICS.

(3) A. Pokorny, P.F. Almeida, The Antibiotic Peptide Daptomycin Functions by Reorganizing the Membrane, J. Membr. Biol. 254 (2021) 97-108. https://doi.org/10.1007/s00232-02100175-0.

(4) L. Robbel, M.A. Marahiel, Daptomycin, a bacterial lipopeptide synthesized by a nonribosomal machinery, J. Biol. Chem. 285 (2010) 2750127508. https://doi.org/10.1074/JBC.R110.128181.

(5) R. Taylor, K. Butt, B. Scott, T. Zhang, J.K. Muraih, E. Mintzer, S. Taylor, M. Palmer, Two successive calcium-dependent transitions mediate membrane binding and oligomerization of daptomycin and the related antibiotic A54145, Biochim. Biophys. Acta - Biomembr. 1858 (2016) 1999-2005. https://doi.org/10.1016/J.BBAMEM.2016.05.020.